# A Comparative Investigation of the Pulmonary Vasodilating Effects of Inhaled NO Gas Therapy and Inhalation of a New Drug Formulation Containing a NO Donor Metabolite (SIN-1A)

**DOI:** 10.3390/ijms25147981

**Published:** 2024-07-22

**Authors:** Attila Oláh, Bálint András Barta, Mihály Ruppert, Alex Ali Sayour, Dávid Nagy, Tímea Bálint, Georgina Viktória Nagy, István Puskás, Lajos Szente, Levente Szőcs, Tamás Sohajda, Endre Zima, Béla Merkely, Tamás Radovits

**Affiliations:** 1Heart and Vascular Center, Semmelweis University, 1122 Budapest, Hungarynagyd1996@gmail.com (D.N.); balint.timea09@gmail.com (T.B.); zima.endre@gmail.com (E.Z.); merkely.study@gmail.com (B.M.);; 2Cyclolab Ltd., 1097 Budapest, Hungarysohtam@gmail.com (T.S.)

**Keywords:** acute pulmonary hypertension, inhalation therapy, pulmonary vasodilation, molsidomine active metabolite, SIN-1A

## Abstract

Numerous research projects focused on the management of acute pulmonary hypertension as Coronavirus Disease 2019 (COVID-19) might lead to hypoxia-induced pulmonary vasoconstriction related to acute respiratory distress syndrome. For that reason, inhalative therapeutic options have been the subject of several clinical trials. In this experimental study, we aimed to examine the hemodynamic impact of the inhalation of the SIN-1A formulation (N-nitroso-N-morpholino-amino-acetonitrile, the unstable active metabolite of molsidomine, stabilized by a cyclodextrin derivative) in a porcine model of acute pulmonary hypertension. Landrace pigs were divided into the following experimental groups: iNO (inhaled nitric oxide, n = 3), SIN-1A-5 (5 mg, n = 3), and SIN-1A-10 (10 mg, n = 3). Parallel insertion of a PiCCO system and a pulmonary artery catheter (Swan-Ganz) was performed for continuous hemodynamic monitoring. The impact of iNO (15 min) and SIN-1A inhalation (30 min) was investigated under physiologic conditions and U46619-induced acute pulmonary hypertension. Mean pulmonary arterial pressure (PAP) was reduced transiently by both substances. SIN-1A-10 had a comparable impact compared to iNO after U46619-induced pulmonary hypertension. PAP and PVR decreased significantly (changes in PAP: −30.1% iNO, −22.1% SIN-1A-5, −31.2% SIN-1A-10). While iNO therapy did not alter the mean arterial pressure (MAP) and systemic vascular resistance (SVR), SIN-1A administration resulted in decreased MAP and SVR values. Consequently, the PVR/SVR ratio was markedly reduced in the iNO group, while SIN-1A did not alter this parameter. The pulmonary vasodilatory impact of inhaled SIN-1A was shown to be dose-dependent. A larger dose of SIN-1A (10 mg) resulted in decreased PAP and PVR in a similar manner to the gold standard iNO therapy. Inhalation of the nebulized solution of the new SIN-1A formulation (stabilized by a cyclodextrin derivative) might be a valuable, effective option where iNO therapy is not available due to dosing difficulties or availability.

## 1. Introduction

Acute pulmonary hypertension is a complex, potentially life-threatening disorder, which may occur both in newborns and adults. Severe pulmonary infection might be associated with pulmonary hypertension as a part of acute lung injury and concomitant hypoxia. Likewise, Coronavirus Disease 2019 (COVID-19) might lead to acute respiratory distress syndrome associated with hypoxia-induced pulmonary vasoconstriction related to generalized inflammation of the endothelium [1,2,3]. Inhaled pulmonary vasodilators, such as nitric oxide (iNO) and prostacyclins, are widely used in the treatment of hospitalized patients with elevated pulmonary arterial pressure, since these agents provide pulmonary vasodilation and thereby improve oxygenation in critically ill patients [4,5,6]. Inhaled pulmonary vasodilators (IPVs) can reduce pulmonary vascular resistance (PVR) and improve right ventricular function with minimal systemic effects. Therefore, inhalation therapy has been the subject of several clinical studies and industrial development projects. 

The standard therapy, NO inhalation, is permitted and reimbursed only in dedicated hospitalized patients and must be controlled by professional medical staff. The main disadvantages of iNO therapy include the extreme instability and dosing difficulties due to rapid inactivation. Additionally, overdosing of iNO might result in methemoglobinemia, toxicity, acute lung injury, and hypoxia as well [7]. To avoid the above-mentioned difficulties, numerous investigations have been launched to compare the effects of parenteral administration of NO donor drugs to apply these therapies for short- and long-term treatment [8,9]. Currently, IPV drug selection always depends on hospital rules and regulations, experience and preferences, and expenses. However, limited and inadequate studies have reported a mortality benefit of IPVs in different patients [10]. Therefore, more research on IPVs is needed to determine management strategies including time, dose, and duration.

An alternative to iNO therapy has been developed by Cyclolab R&D Ltd. (Budapest, Hungary). SIN-1A (N-nitroso-N-morpholino-amino-acetonitrile), the unstable active metabolite of the orally administered prodrug molsidomine and linsidomine (SIN-1), has been stabilized by a cyclodextrin derivative in a new drug formulation. This cyclodextrin derivative might facilitate the administration and proper dosing of SIN-1A by inhalation. In this experimental study, we aimed to investigate the hemodynamic effect of inhalative SIN-1A administration under physiological and pathological conditions in a large animal model and compare its efficacy to the standard iNO inhalation therapy. According to our hypothesis, SIN-1A administration might be an effective option to resolve acute pulmonary hypertension with the advantage of ease of administration.

## 2. Results 

### 2.1. Pharmacodynamics

SIN-1A administration showed a similar, rapid effect on pulmonary artery pressure as iNO inhalation (Figure 1). The greatest relative change in pulmonary artery pressure values was reached after approximately ten minutes and remained relatively unchanged during further administration. Larger doses of SIN-1A (SIN-1A-10) were comparable to iNO-related PAP alterations. 

### 2.2. Hemodynamic Alterations after Inhalation Therapy—Physiological Conditions

Baseline hemodynamic data were compared to measurements collected after inhalational therapy of each drug (Table 1 and Figure 2). PAP and PVR showed a tendentious decrease due to all of these drugs and SIN-1A-10 had a similar effect as iNO. While iNO did not influence SAP and SVR, SIN-1A administration was associated with decreased SAP and SVR values. Moreover, this effect seemed to be dose-dependent: higher doses of SIN-1A had a more pronounced effect (Figure 2). As a result of these alterations, the PVR/SVR ratio decreased in the iNO group, while it remained unchanged in the SIN-1A-5 and SIN-1A-10 groups. HR and CO were not significantly altered.

### 2.3. Hemodynamic Alterations after Inhalation Therapy—Pulmonary Hypertension (U46619)

The observed alterations after inhalation therapy were similar under pathological conditions (Table 1 and Figure 2). iNO markedly decreased the PAP and PVR values. The SIN-1A-10 dose had a similar impact on PAP as iNO, while SIN-1A-5 inhalation was associated with a moderate decrement (~20%). Both SIN-1A doses were associated with decreased PVR, although the larger dose showed only a strong tendency. SIN-1A-5 and SIN-1A-10 also reduced systemic blood pressure and vascular resistance (SAP and SVR), while these parameters remained unchanged in the iNO group. Consequently, iNO markedly decreased the PVR/SVR ratio, showing the strong selectivity of its vasodilating effect on the pulmonary circulation. In contrast, SIN-1A was associated with an unchanged PVR/SVR ratio. We could not detect any significant alterations in CO and HR.

### 2.4. Other Parameters

The core temperature and the parameters of blood gas analysis were considered physiological throughout our experiments. Inhalation therapies did not significantly influence platelet aggregation (using ADP reagent, Table 2) under physiological conditions. SIN-1A inhalation tended to decrease the AUC values in the case of U46619-induced pulmonary hypertension, indicating a possible beneficial inhibitory effect on platelet aggregation under pathological conditions.

## 3. Discussion 

In this study, we provided a direct comparison of the short-term pulmonary and systemic effects of the new inhaled SIN-1A formulation and compared it to the gold standard pulmonary vasodilator iNO therapy in a porcine model of acute pulmonary hypertension. According to our data, SIN-1A might be an effective therapeutic option in acute pulmonary hypertension. 

Pulmonary hypertension is a characteristic and important feature of acute respiratory distress syndrome (ARDS). The early phase of ARDS involves the pathological processes of pulmonary vasoconstriction, thromboembolism of the small and large vessels, and lung interstitial edema [11]. These pathophysiologic mechanisms are also associated with severe COVID-19 pneumonia, promoting the sequence of acute pulmonary hypertension [2]. Not only the micro- and macro-embolization in the pulmonary circulation, but also the generalized injury of the endothelium plays a pivotal role in pulmonary injury by losing the active paracrine, endocrine, and autocrine vasodilative function of the vascular endothelium [12]. The production of vasodilators, such as nitric oxide and prostacyclin, might be severely impaired, promoting vasoconstriction of the pulmonary vessels. These factors set the stage for the development of acute pulmonary hypertension and associated problems, such as right ventricular strain and failure [2].

We found similar pharmacodynamics of SIN-1A and iNO (Figure 1). Compared with other routes of drug administration, inhalation therapy delivers medication directly to the lung, enabling higher pulmonary drug concentrations and less systemic adverse effects. Indeed, the velocity of the effect is comparable to other inhaled pulmonary vasodilators such as iloprost, a prostacyclin analog [13]. The route of inhalation has a number of attractive features for the treatment of pulmonary hypertension, including the delivery of the drug directly to the target organ, thus enhancing pulmonary specificity and reducing systemic adverse effects [14]. Intravenous vasodilator agents might lead to an increase in intrapulmonary shunting and systemic hypotension, which can limit their therapeutic use [11]. Nitric oxide as an endogenous vasodilatory substance is of particular interest in pulmonary hypertension, considering its selective pulmonary action [15]. Although iNO therapy has not been associated with improved survival rates in adult intensive care patients, the application of iNO inhalation can serve as a salvage therapy for ARDS in adults, as it temporarily improves arterial oxygenation [16]. SIN-1A (3-morpholino-syndnonimine) is the active metabolite of the orally administered prodrug molsidomine stabilized by cyclodextrin. Therefore, its stability allows us to provide inhalation therapy without a complicated delivery system. In our experiment, iNO therapy was provided as the standard dose for therapy (20 ppm) that might be associated with the optimized benefit–risk ratio by providing significant hemodynamic impact with less probability of adverse effects [17,18,19]. 

Inhalation of NO was associated with a significant reduction in PAP and PVR (approximately 30–40% decrease in pressure) and its effect was comparable to other studies, where iNO therapy was provided in pathological conditions in human and animal models [14,19,20]. With the potential toxicity, rebound phenomenon, high cost, and unsatisfactory response rate associated with iNO, alternative treatment options have been tested. Inhaled prostacyclin showed similar efficacy compared to iNO therapy [21]. SIN-1A therapy was also associated with a significant reduction in PAP. Oral administration of the NO donor molsidomine might attenuate hypoxia-related PH syndrome by enhancing the NO-cGMP pathway in experimental settings and in human cases [19,22,23]. Therefore, inhalation of molsidomine-derived pharmacological agents might result in an effective local treatment in alveoli, where pulmonary vasoconstriction is the primary cause of ventilation–perfusion mismatch. We could also observe the dose-dependent impact of SIN-1A: while 5 mg could not lead to the same pronounced effect as iNO, SIN-1A-10 therapy was associated with a similar impact on the pulmonary circulation (Table 1 and Figure 2). This is an important feature as other promising drugs could not reach the same effect in previous studies as iNO [24]. We should also mention that SIN-1A-10 showed similar pharmacodynamics to iNO regarding pulmonary resistance, reaching its maximal effect after 15 min (Figure 1). The lack of delay in effect might be a crucial characteristic in the setting of acute PH, especially in the case of ARDS caused by fulminant pulmonary infection.

Although iNO markedly decreased pulmonary resistance, the effect on the systemic circulation was not significant: both SAP and PVR remained unchanged in this group, regardless of the physiological or pathological pulmonary vascular state (Figure 1 and Figure 2). This is in accordance with previous studies that investigated the effect of NO inhalation and might be associated with the rapid metabolization due to the instability of the molecule [13,14]. In contrast to the standard therapy, SIN-1A showed a dose-dependent systemic vasodilator effect: both SAP and SVR decreased after inhalation of the lower and higher dose of SIN-1A (Table 1 and Figure 2). This is a common side effect of drugs tested in the treatment of acute pulmonary hypertension [25]. Certainly, the NO donor molsidomine might provide systemic antihypertensive effects through its active metabolite [26]. Also, the current data suggest that the dissociation of NO might show a delayed release and consequent impact in the current molecular formula, where cyclodextrin was used to stabilize the active metabolite.

Although the properties of the ideal pulmonary vasodilator include selectivity for the pulmonary circulation, systemic vasodilatation might be effectively counterbalanced by selective systemic vasopressors (such as alpha-adrenergic agonist pharmacological agents). On the other hand, SIN-1A might be favorable in cases where systemic and pulmonary hypertension exist simultaneously [27]. Indeed, systemic arterial hypertension is associated with increased mortality and severity in COVID-19 pneumonia [28]. Above all, the ease of administration of SIN-1A (nebulization) and its efficacy in relieving pulmonary hypertension makes it a promising new formulation to treat acute pulmonary hypertension.

NO donor drugs might alter platelet function; therefore, we performed platelet aggregometry testing (Table 2). In accordance with previous results, SIN-1A therapy was associated with a tendency toward improved platelet aggregation, suggesting a favorable effect on platelet function [29]. 

We should also mention some limitations: the small number of animals due to ethical considerations and the lack of inflammation features in our TXA2 agonist-induced acute pulmonary hypertension model are the main disadvantages of these experiments. On the other hand, the detailed, precise hemodynamic measurements in large animals might turn them into a valuable study.

## 4. Methods and Materials

### 4.1. Study Groups

Male and female landrace pigs (body weight: 20–30 kg) provided by the Research Institute for Animal Breeding, Nutrition and Meat Science of the Hungarian University of Agriculture and Life Sciences (Herceghalom, Hungary) were randomized into experimental groups (Figure 3). 

The investigation conformed to the EU Directive 2010/63/EU and was in accordance with ARRIVE guidelines. The experiments were approved by the Ethical Committee of Hungary for Animal Experimentation (permission number: PE/EA/477-4/2021).

### 4.2. Animal Preparation

The animals were sedated by an im. injection of 25 mg/kgBW ketamine (50 mg/mL, Gedeon Richter Plc. Budapest, Hungary) and 0.3 mg/kgBW midazolam (15 mg/3 mL, Kalceks AS, Riga, Latvia) and were carefully transported into the laboratory. Anesthesia was induced by a 2.5 mg/kgBW iv. propofol (10 mg/mL, Fresenius Kabi, Bad Homburg, Hessen, Germany) injection into the ear vein, followed by orotracheal intubation with an endotracheal tube after which controlled ventilation was applied (Dräger Primus, Drägerwerk AG & Co., Ltd., Lübeck, Schleswig-Holstein, Germany). Baseline parameters of ventilation (tidal volume: 10–12 mL/kgBW, frequency: 15–18) were adjusted according to arterial blood gas test results (Cobas b221, Roche, Basel, Switzerland) to maintain arterial partial carbon dioxide pressure levels at 40 mmHg, while the fraction of inspired oxygen (FiO_2_) was maintained at 50% for the entire duration of the experiment. Anesthesia was maintained with 10 mg/kgBW/h continuous propofol infusion, while analgesia was provided by an im. injection of 0.3 mg/kgBW butorphanol (10 mg/mL, Nalgosed, Bioveta, Czech Republic). Basic intravenous volume substitution was carried out via the left external jugular vein with Ringer’s solution at a rate of 1 mL/min/kgBW. To prevent thromboembolism, the animals were heparinized with 5000 IU Na-heparin iv.

### 4.3. Animal Monitoring and Hemodynamic Data

Major veins and arteries were surgically prepared and cannulated as follows: the left femoral vein was used for blood sampling; the left external jugular vein was used for volume substitution/drug administration, the measurement of central venous pressure (CVP), and the injection of a cold saline solution to determine cardiac output (CO) according to the thermodilution method using a PiCCO system (PULSION Medical Systems PiCCO2–PC8500, Munich, Bavaria, Germany); the left femoral artery was used to measure systemic arterial pressure and to obtain blood samples for arterial blood gas tests; the right external jugular vein was used to introduce a Swan–Ganz catheter (AI-07124, 5 Fr. 110 cm, Arrow Internat Inc., Reading, PA, USA) into the pulmonary artery to measure pulmonary arterial pressure and pulmonary capillary wedge pressure. 

Heart rate (HR), mean systemic arterial pressure (SAP), mean pulmonary arterial pressure (PAP), pulmonary capillary wedge pressure (PCWP), and ECG data were collected using instruments from Pulsion Medical Systems, and Powerlab, AD-Instruments (Castle Hill, Australia). Pulmonary vascular resistance (PVR) and systemic vascular resistance (SVR) were calculated as PVR = (PAP_mean_ − PCWP)/CO and SVR = (SAP_mean_ − CVP)/CO, respectively. Hemoglobin oxygen saturation (Sat%) was determined with a sensor placed on the tail, while temperature (Temp) was continuously measured with a rectal probe.

### 4.4. Experimental Protocol

The experimental design is depicted in Figure 2. After anesthesia, intubation and instrumentation baseline parameters (SAP, PAP, CVP, PCWP, CO, HR, Sat%, Temp) were registered following a 15 min long stabilization period (BASELINE). To test the hemodynamic effect of the vasodilatory agents under physiologic conditions, inhalation of either iNO (20 ppm, 15–30 min, until a stable plateau was reached) or SIN-1A (5 or 10 mg for 30 min) depending on the experimental group was applied (BASELINE + INH). This was followed by a 30 min wash-out/equilibration period. To compare hemodynamic responses to iNO and the aforementioned two doses of SIN-1A under pathologic circumstances, pulmonary hypertension was induced by continuous infusion of U46619 (60–120 ng/kg/min) until achieving the target mean pulmonary arterial pressure (mPAP, 35–45 mmHg). After achieving a steady state, hemodynamic parameters were recorded (U46619) and another 15–30 min long inhalation administration of iNO or SIN1 was initiated, followed by repeated registration of hemodynamic data (U46619 + INH).

### 4.5. SIN-1A Drug Formula and Administration

In the current experiments, a stabilized form of SIN-1A was utilized to test its effect on the pulmonary and systemic circulation (Figure 4). Molsidomine is a prodrug that undergoes first-pass metabolism in the liver, and thus, linsidomine (SIN-1) is formed. SIN-1 is further decomposed nonenzymatically to SIN-1A, then to SIN-1^•+^, which spontaneously releases NO, and finally, SIN-1C is formed. The used cyclodextrin (2-hydroxypropyl)-beta-cyclodextrin (HPBCD) is an accepted excipient by the European Medicines Agency (EMA) and by the U.S. Food and Drug Administration (FDA) as well. According to the EMA guideline for cyclodextrins, HPBCD at high doses can increase nasal and pulmonary drug permeability, and solutions less than 10% do not induce tissue damage in rats. Currently, there is no approved inhalational final dosage form that contains HPBCD. SIN-1A at 20 mg or 10 mg was freshly dissolved in 3 mL of a special solvent supplied by Cyclolab. During a 30 min inhalation period, 1.5 mL of the solution (10 mg SIN-1A-10 group or 5 mg SIN-1A-5 group, respectively) was introduced to the ventilator circuit using a standard clinical nebulizer system (Aerogen Pro-X, Galway, Ireland).

### 4.6. Other Drugs

The thromboxane A2-mimetic 9,11-dideoxy-9α, 11α-methanoepoxy PGF2α, U46619 (Cayman Chemical, Ann Arbor, MI, USA), was supplied in methyl acetate and diluted in 96% ethanol to obtain stock solutions of 1 mg/mL, which were diluted in Ringer’s solution to a final concentration of 3 µg/mL. iNO was supplied from a tank (800 parts per million [ppm] of nitrogen) and introduced to the inspiratory limb of the ventilator circuit through a nitric oxide delivery and monitoring system (NOxBOX, Kent, UK) to mix NO (20 ppm) with inspired gas. A portion of inhaled NO was continuously measured to prevent fluctuations in concentration.

### 4.7. Platelet Aggregation

At 4 time points (BASELINE, BASELINE + INH, U46619, U46619 + INH), 2.7 mL venous blood samples were taken and filled in hirudin-coated tubes and remained unaffected for 60 min. Platelet inhibition was quantified with a Multiplate^®^ Analyzer (Roche, Basel, Switzerland). The test was conducted as instructed by the manufacturer. Briefly, 300 μL blood was transferred into three measuring cells, mixed with 300 μL of 37 °C warm isotonic sodium chloride, and incubated for a three-minute-long period. A total of 20 μL of the reagent (adenosine diphosphate (ADP) or arachidonic acid (AA)) was pipetted into these samples. Six minutes of measurement time per each test (ADP test and ASPI test) were allowed. The intensity of platelet aggregation and the aggregation velocity are indicated in arbitrary units (area under the curve [AUC]).

### 4.8. Statistical Analysis

The data are represented as the mean ± standard error of the mean (SEM).

Responses to the administration of inhalation drugs (before administration and after steady-state effect) were evaluated with paired *t*-tests. We compared their effect in a pairwise mode under physiological conditions (BASELINE vs. BASELINE + INH) and during induced acute pulmonary hypertension (U46619 vs. U46619 + INH). A value of *p* < 0.05 was considered statistically significant.

## 5. Conclusions

In the present proof-of-concept study, inhalation of the new nebulized drug formulation containing the NO donor metabolite SIN-1A stabilized by HPBCD rapidly and effectively reduced pulmonary arterial pressure in pharmacologically induced acute pulmonary hypertension by decreasing PAP and PVR values. The pulmonary vasodilator effect was dose-dependent, and the impact of larger doses of SIN-1A was comparable to that of iNO. In contrast to iNO, the vasodilator effect of SIN-1A was not limited to the pulmonary circulation; systemic vasodilation (decreased SAP and SVR) could also be documented. SIN-1A might be a valuable therapeutic option where iNO administration could encounter problems due to dosing difficulties or availability.

## Figures and Tables

**Figure 1 ijms-25-07981-f001:**
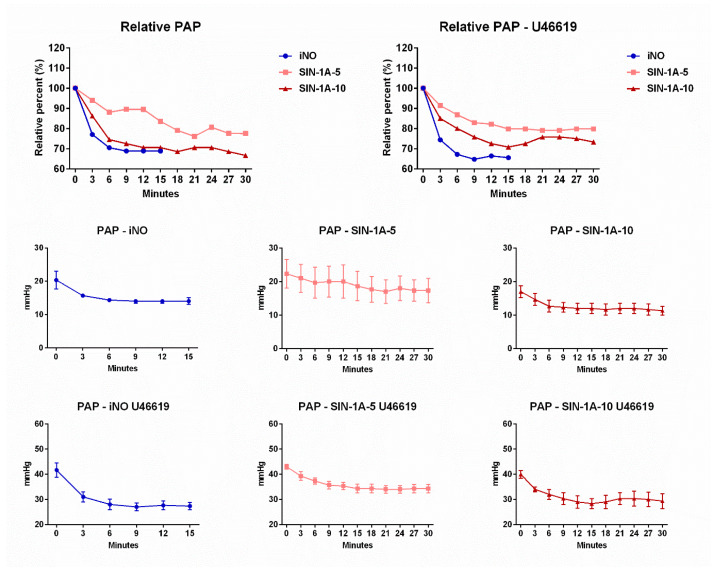
Pharmacodynamics of inhaled nitric oxide (iNO) and inhaled SIN-1A formulation at different doses (SIN-1A-5 5 mg, SIN-1A-10 10 mg). PAP: pulmonary artery pressure.

**Figure 2 ijms-25-07981-f002:**
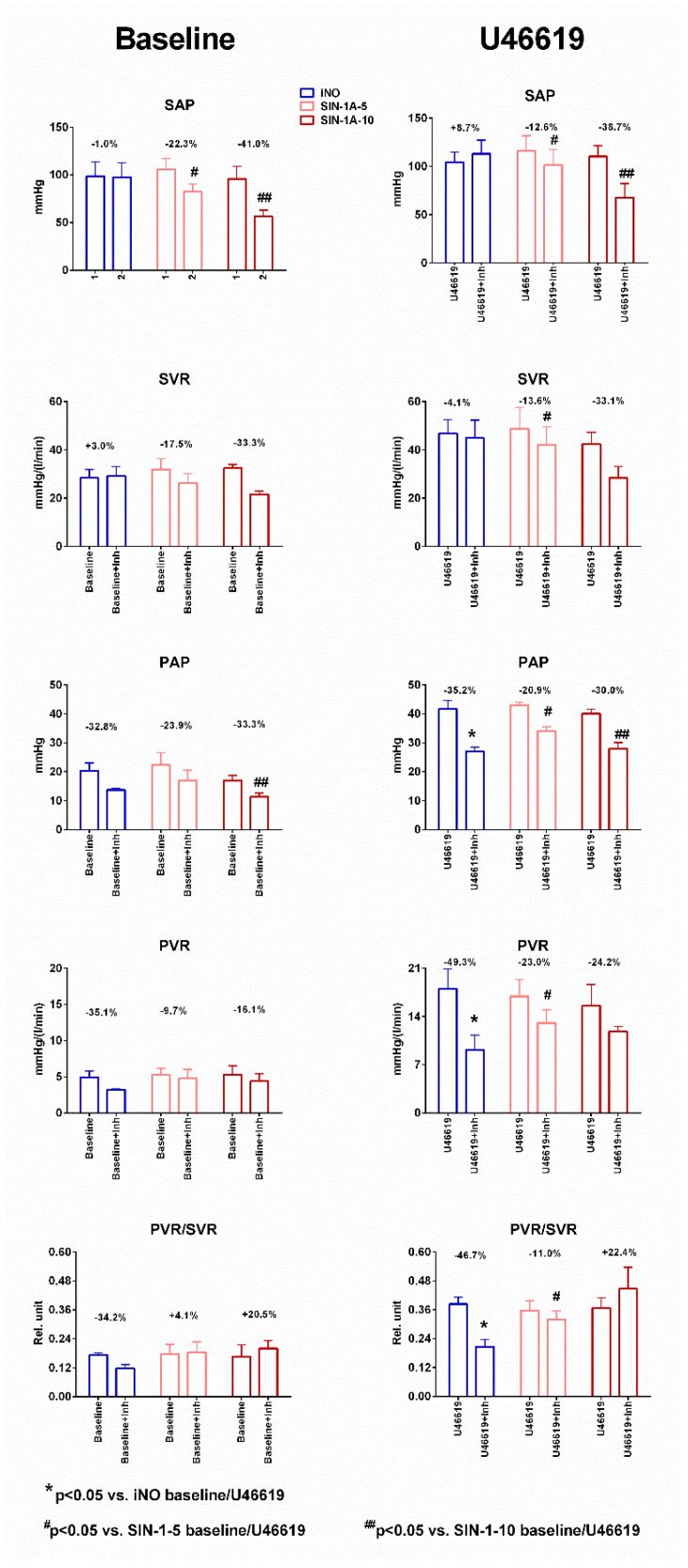
Alterations in systemic and pulmonary circulation during administration of inhaled nitric oxide (iNO) and inhaled SIN-1A formulation at different doses (SIN-1-5 5 mg, SIN-1-10 10 mg). PAP: pulmonary artery pressure, PVR: pulmonary vascular resistance, SAP: systemic arterial pressure, SVR: systemic vascular resistance. * *p* < 0.05 vs. iNO baseline vs. baseline + inhalation or U46619 vs. U46619 + inhalation. ^#^
*p* < 0.05 vs. SIN-1A-5 baseline vs. baseline + inhalation or U46619 vs. U46619 + inhalation. ^##^
*p* < 0.05 vs. SIN-1A-10 baseline vs. baseline + inhalation or U46619 vs. U46619 + inhalation.

**Figure 3 ijms-25-07981-f003:**
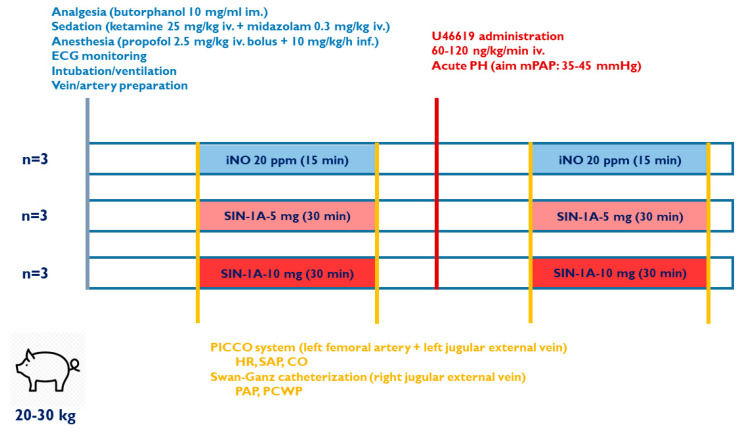
Experimental setting: preparation steps, hemodynamic monitoring, and induction of acute pulmonary hypertension (PH). Dosing of inhaled nitric oxide (iNO) and inhaled SIN-1A formulation at different doses (SIN-1A-5 5 mg, SIN-1A-10 10 mg inhalation). CO: cardiac output, HR: heart rate, PAP: pulmonary artery pressure, PCWP: pulmonary wedge pressure, SAP: systemic arterial pressure. iNO, n = 3; SIN-1A 5 mg (SIN-1A-5), n = 3; SIN-1A 10 mg (SIN-1A-10), n = 3.

**Figure 4 ijms-25-07981-f004:**
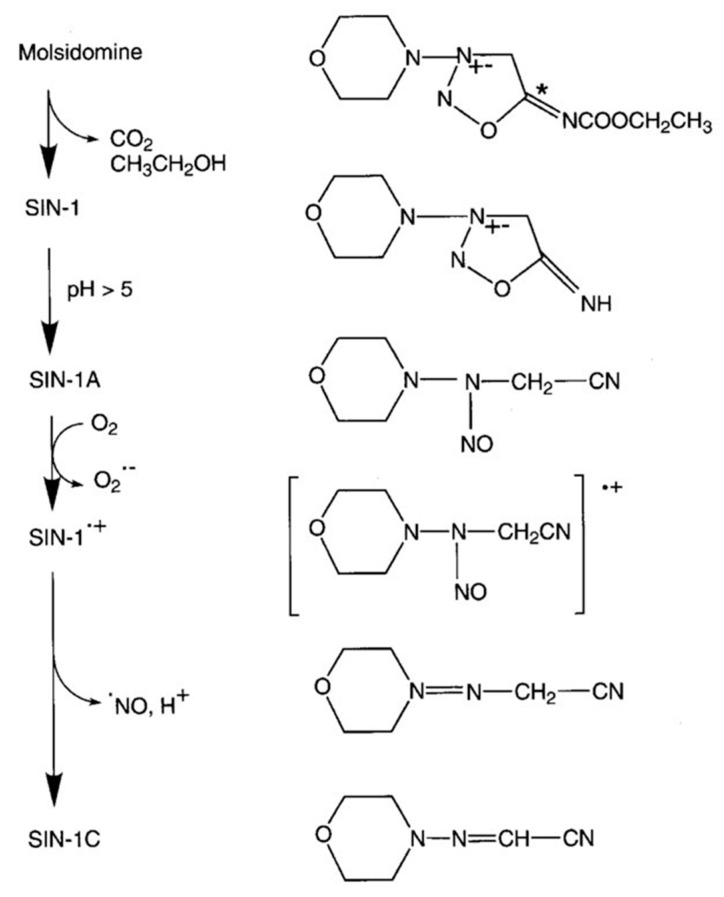
Metabolism and NO release of molsidomine. * is a chemical common sign.

**Table 1 ijms-25-07981-t001:** Alterations in hemodynamic parameters during administration of inhaled nitric oxide (iNO) and inhaled SIN-1A formulation at different doses (SIN-1A-5 5 mg, SIN-1A-10 10 mg). CO: cardiac output, CVP: central venous pressure, HR: heart rate, mPAP: mean pulmonary artery pressure, PCWP: pulmonary wedge pressure, PVR: pulmonary vascular resistance, SAP: systemic arterial pressure, SVR: systemic vascular resistance. * *p* < 0.05 vs. iNO baseline vs. baseline + inhalation or U46619 vs. U46619 + inhalation. ^#^
*p* < 0.05 vs. SIN-1A-5 baseline vs. baseline + inhalation or U46619 vs. U46619 + inhalation. ^##^
*p* < 0.05 vs. SIN-1A-10 baseline vs. baseline + inhalation or U46619 vs. U46619 + inhalation.

		Baseline	Baseline + Inhalation	U46619	U46619 + Inhalation
iNO	PAP (mmHg)	20.3 ± 2.7	14.0 ± 1.0	41.7 ± 2.8	27.7 ± 1.8 *
	PVR (mmHg/L/min)	5.5 ± 0.8	3.4 ± 0.1	20.1 ± 2.8	10.8 ± 2.2 *
	PCWP (mmHg)	3.7 ± 0.9	3.3 ± 0.7	3.8 ± 1.4	5.3 ± 1.2 *
	SAP (mmHg)	98.7 ± 15.0	97.7 ± 14.8	104.0 ± 10.8	113.0 ± 14.2
	SVR (mmHg/L/min)	28.4 ± 3.5	29.2 ± 3.9	46.8 ± 5.7	44.8 ± 7.4
	CVP (mmHg)	3.0 ± 1.5	3.0 ± 1.5	4.7 ± 2.3	4.3 ± 1.3
	CO (L/min)	3.3 ± 0.3	3.2 ± 0.2	2.1 ± 0.2	2.5 ± 0.2
	HR (beats/min)	110 ± 9	100 ± 6	127 ± 8	108 ± 12
	PVR/SVR	0.17 ± 0.01	0.10 ± 0.01	0.38 ± 0.03	0.21 ± 0.03 *
SIN-1A-5	PAP (mmHg)	22.3 ± 4.3	17.5 ± 3.5	43.0 ± 1.0	34.3 ± 1.7 ^#^
	PVR (mmHg/L/min)	5.3 ± 0.9	5.1 ± 1.3	17.0 ± 2.4	13.2 ± 1.9 ^#^
	PCWP (mmHg)	4.7 ± 0.7	2.3 ± 0.9	4.7 ± 0.7	3.7 ± 0.7
	SAP (mmHg)	106.0 ± 11.1	82.3 ± 7.9 ^#^	116.3 ± 15.4	101.7 ± 16.0 ^#^
	SVR (mmHg/L/min)	31.9 ± 4.5	26.3 ± 3.9	48.7 ± 8.9	42.1 ± 4.8 ^#^
	CVP (mmHg)	3.3 ± 0.9	2.3 ± 1.5	6.0 ± 0.6	3.0 ± 0.6 ^#^
	CO (L/min)	3.3 ± 0.3	3.1 ± 0.1	2.3 ± 0.2	2.4 ± 0.2
	HR (beats/min)	90 ± 5	98 ± 10	109 ± 6	118 ± 10
	PVR/SVR	0.17 ± 0.04	0.19 ± 0.04	0.36 ± 0.04	0.32 ± 0.03 ^#^
SIN-1A-10	PAP (mmHg)	17.0 ± 1.7	11.3 ± 1.3 ^##^	40.0 ± 1.5	28.3 ± 2.0 ^##^
	PVR (mmHg/L/min)	5.3 ± 1.3	4.4 ± 1.0	15.6 ± 3.1	12.0 ± 0.6
	PCWP (mmHg)	2.7 ± 1.7	1.3 ± 0.3	2.3 ± 0.3	1.7 ± 0.7
	SAP (mmHg)	96.0 ± 12.7	56.7 ± 6.3 ^##^	110.3 ± 11.3	67.7 ± 14.6 ^##^
	SVR (mmHg/L/min)	32.5 ± 1.5	21.6 ± 1.3	42.4 ± 4.9	28.3 ± 4.8
	CVP (mmHg)	3.7 ± 2.2	2.7 ± 1.7	4.7 ± 1.5	3.7 ± 1.7
	CO (L/min)	2.8 ± 0.2	2.5 ± 0.3	2.6 ± 0.4	2.2 ± 0.2
	HR (beats/min)	90 ± 13	110 ± 9	137 ± 24	134 ± 24
	PVR/SVR	0.17 ± 0.05	0.20 ± 0.03	0.36 ± 0.04	0.45 ± 0.08

**Table 2 ijms-25-07981-t002:** Platelet aggregation results during administration of inhaled nitric oxide (iNO) and inhaled SIN-1A formulation at different doses (SIN-1A-5 5 mg, SIN-1A-10 10 mg). AUC: area under curve.

	Baseline (AUC)	Baseline + Inhalation (AUC)	U46619 (AUC)	U46619 + Inhalation (AUC)
iNO	50 ± 13	54 ± 15	49 ± 19	52 ± 20
SIN-1A-5	66 ± 19	62 ± 12	49 ± 6	44 ± 8
SIN-1A-10	70 ± 19	75 ± 22	52 ± 23	45 ± 23

## Data Availability

The datasets used and/or analyzed during the current study are available from the corresponding author upon reasonable request.

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
