# Peer review of "A Comparative Investigation of the Pulmonary Vasodilating Effects of Inhaled NO Gas Therapy and Inhalation of a New Drug Formulation Containing a NO Donor Metabolite (SIN-1A)"

_ijms, 2024, doi:10.3390/ijms25147981_

Round 1
Reviewer 1 Report
Comments and Suggestions for Authors
The topic is interesting and the paper is quite well written. The article covers a very interesting and current topic. Nevertheless, in my opinion, some parts need to be improved, I have some comments:
1) Background and purpose: Numerous research projects focused on the management of acute pulmonary hypertension as Coronavirus Disease 2019 (COVID-19) might lead to hypoxia-induced pulmonary vasoconstriction related to acute respiratory distress syndrome. For that reason, inhalative therapheutic options have been the subject of several clinical trials. In this experimental study we aimed at examining the hemodynamic impact of inhalative administration of the SIN-1A formulation (N-nitroso-N-morpholino-amino-acetonitrile, the unstable active metabolite of molsidomine, stabilized by a cyclodextrin derivative) in a porcine model of acute pulmonary hypertension. Experimental Approach: Landrace pigs were divided into the following experimental groups: iNO (inhaled nitric-oxide,n=3), SIN-1A-5 (5 mg,n=3), SIN-1A-10 (10 mg,n=3). Parallel insertion of a PiCCO system and a pulmonary artery catheter (Swan-Ganz) were performed for continuous hemodynamic monitorization. Acute pulmonary hypertension was induced by the U46619, a thromboxane-receptor agonist. The impact of iNO (15min) and SIN-1A inhalation (30 min) was investigated under physiologic and pathological circumstances. Key Results: Inhalation therapy resulted in similar alterations both under physiological and pathological conditions. Pulmonary arterial pressure (PAP) was reduced by all of these substances, SIN-1A-10 had comparable impact compared to iNO (physiological condition: -36.1% iNO, -23.8% SIN-1A-5, -35.5% SIN-1A-10; U46619-induced pulmonary hypertension: -30.1% iNO, -22.1% SIN-1A-5, -31.2% SIN-1A-10). Inhaled NO therapy did not alter the mean arterial pressure (MAP) and systemic vascular resistance (SVR) values. At the same time SIN-1A administration resulted in decreased MAP and SVR values. Calculated pulmonary vascular resistance (PVR) was decreased by iNO to greater extent compared to SIN-1A inhalation. Consequently, the PVR/SVR ratio reduced markedly in the iNO group, while it was stepwisely incremented in the SIN-1A-5 and SIN-1A-10 group. Conclusions and Implications: The pulmonary vasodilator impact of inhaled SIN-1A has shown to be dose-dependent, and was comparable to that of iNO. Inhalation of the nebulized solution of the new SIN-1A formulation might be a valuable, effective option, where iNO therapy is not available. Please, the abstract is quite rumbling and difficult to read. I suggest to clarify the results to support the conclusions.
2) Abstract. Conclusions and Implications: The pulmonary vasodilator impact of inhaled SIN-1A has shown to be dose-dependent, and was comparable to that of iNO. Inhalation of the nebulized solution of the new SIN-1A formulation might be a valuable, effective option, where iNO therapy is not available. Abstract might be beneficial to include a sentence that briefly summarizes the key findings of the study. This can provide readers with a quick overview of the research.
3) An alternative to iNO therapy has been developed by Cyclolab R&D Ltd.(Budapest, 60 Hungary): SIN-1A (N-nitroso-N-morpholino-amino-acetonitrile), the unstable active me- 61 tabolite of the orally administered prodrug molsidomine and linsidomine (SIN-1), has 62 been stabilized by a cyclodextrin derivative in a new drug formulation. This cyclodextrine 63 might facilitate the administration and proper dosing of SIN-1A by inhalation. In this ex- 64 perimental study we aimed at investigating the hemodynamic effect of inhalative SIN-1A 65 administration during physiological and pathological conditions in a large animal model 66 and compared its efficacy to the standard iNO inhalation therapy. Please underline the aim of the study and the novelty of this work.
4) Statistical analysis 177 Data are represented as the mean ± standard error of the mean (SEM). Responses to 178 administration of inhalation drugs were assessed with paired t-tests (BASELINE vs. 179 BASELINE+INH; U46619 vs. U46619+INH. p<0.05 was considered as statistically signifi- 180 cant. Please, underline the statistical tests used to evaluate the data for supporing the conclusions.
5) Results. Improve this section and support the most important results by the statistcally significant values.
6) Discussion 247 In this study, we provided a direct comparison of short-term pulmonary and sys- 248 temic effects of the new inhaled SIN-1A formulation and compared it to the gold standard 249 pulmonary vasodilator iNO therapy in a porcine model of acute pulmonary hypertension. 250 According to our data, SIN-1A might be an effective therapeutic option in acute pulmo- 251 nary hypertension. The discussion section needs to be improved. It is necessary to clarify the results obtained and compare them with published literature.
Comments on the Quality of English Language
Minor changes of English language are required
Author Response
Reviewer #1
We would like to thank the careful evaluation of Reviewer #1 and the helpful and constructive suggestions. A point-by-point response has been provided below.
1-2) Please, the abstract is quite rumbling and difficult to read. I suggest to clarify the results to support the conclusions. Abstract might be beneficial to include a sentence that briefly summarizes the key findings of the study. This can provide readers with a quick overview of the research.
According to the request of the reviewer we have re-written the results and conclusions parts of our abstract (line 25-36). The results section has been significantly altered. We also emphasized the current finding for a better overview.
3) Please underline the aim of the study and the novelty of this work.
We modified the introduction part regarding this helpful suggestion of Reviewer #1. We added a sentence at the end of the phrased paragraph that clarifies our aim (Line 66-67).
4) Please, underline the statistical tests used to evaluate the data for supporting the conclusions.
The description of the statistic evaluation has been widened to clarify the interpretation of the data. We would also add that the statistical evaluation is very limited using such small number of animals.
5) Results. Improve this section and support the most important results by the statistically significant values.
Regarding the suggestion of Reviewer #1 and #2, we altered the results part to obtain a more clear description. We changed the Figure of hemodynamic parameters to a table, which shows absolute values of the hemodynamic parameters and might provide a more clear data. Also, we tried to emphasize the significant alterations.
6) The discussion section needs to be improved. It is necessary to clarify the results obtained and compare them with published literature.
We would like to thank for this constructive suggestion for the Reviewer. We tried to improve the discussion by inserting additional citations. These articles include valuable data about treating acute pulmonary hypertension and form a basis for comparison.
Reviewer 2 Report
Comments and Suggestions for Authors
This study explores the hemodynamic effects of the inhalative administration of SIN-1A in a porcine model to expand therapeutic options potentially. The manuscript could be significantly strengthened for submission by addressing the following points.
1. More detail could be added about previous findings or gaps this study aims to fill regarding SIN-1A use in pulmonary hypertension.
2. Results are presented as relative changes, which might be difficult to interpret without baseline values or absolute changes. Consider presenting both to provide a clearer picture of the effects.
3. Discuss broader implications of the findings for clinical practice, especially in settings where iNO is unavailable.
4. The conclusion could be more concise and focused on the main findings rather than reiterating methodological details.
Comments on the Quality of English LanguageMinor editing of English Language required.
Author Response
Reviewer #2
We would like to thank the careful evaluation of Reviewer #2 and the helpful and constructive suggestions. A point-by-point response has been provided below.
- More detail could be added about previous findings or gaps this study aims to fill regarding SIN-1A use in pulmonary hypertension.
According to this suggestion we inserted few sentences to the first and second paragraph of the introduction and emphasized the need for research about inhaled pulmonary vasodilators (line 41-65).
- Results are presented as relative changes, which might be difficult to interpret without baseline values or absolute changes. Consider presenting both to provide a clearer picture of the effects.
Regarding the suggestion of Reviewer #1 and #2, we altered the results part to obtain a more clear description. We changed the previous figure of hemodynamic parameters to a table (Table 1.), which shows absolute values of the hemodynamic parameters and might provide a more clear data.
- Discuss broader implications of the findings for clinical practice, especially in settings where iNO is unavailable.
Accordingly, we have tried to improve the introduction and discussion by insertion of additional citations (including studies those involved inhalation of iloprost and prostacyclins). These studies also form a basis for direct comparison to the effect of SIN-1A and iNO inhalation.
- The conclusion could be more concise and focused on the main findings rather than reiterating methodological details.
Based on this concern of Reviewer#2 we modified the conclusion to focus more on the current findings. We also involved parameters and tried to emphasize the relevance of our results (line 336-344).
Round 2
Reviewer 1 Report
Comments and Suggestions for Authors
Please, underline in the manuscript all the changes, I suggest to underline the text or use the red colour for the revised part

Author Response
Please, underline in the manuscript all the changes, I suggest to underline the text or use the red colour for the revised part.
We would apologize for the lack of manuscript with track changes, however it was not requested in the file list. Therefore we hereby attach the "track changes" version. We hope that the Reviewer can follow-up the alterations made by us during the revision process.

Round 3
Reviewer 1 Report
Comments and Suggestions for Authors
The manuscript has been improved, as requested. No further comments
Comments on the Quality of English LanguageMinor changes of English language are required